# Fads2b Plays a Dominant Role in ∆6/∆5 Desaturation Activities Compared with Fads2a in Common Carp (*Cyprinus carpio*)

**DOI:** 10.3390/ijms241310638

**Published:** 2023-06-26

**Authors:** Ran Zhao, Chen-Ru Yang, Ya-Xin Wang, Zi-Ming Xu, Shang-Qi Li, Jin-Cheng Li, Xiao-Qing Sun, Hong-Wei Wang, Qi Wang, Yan Zhang, Jiong-Tang Li

**Affiliations:** Key Laboratory of Aquatic Genomics, Ministry of Agriculture and Rural Affairs and Beijing Key Laboratory of Fishery Biotechnology, Chinese Academy of Fishery Sciences, Beijing 100141, China; zhaoran@cafs.ac.cn (R.Z.); ycrrlucky@163.com (C.-R.Y.); wangyaxin930629@163.com (Y.-X.W.); xuziming0916@163.com (Z.-M.X.); lisq@cafs.ac.cn (S.-Q.L.); lijc@cafs.ac.cn (J.-C.L.); sunxiaoqing@cafs.ac.cn (X.-Q.S.); wanghongwei@cafs.ac.cn (H.-W.W.); wangqi@cafs.ac.cn (Q.W.)

**Keywords:** common carp, fatty acid desaturase 2, highly unsaturated fatty acids, desaturation activity, dominant expression

## Abstract

Highly unsaturated fatty acids (HUFAs) are essential for mammalian health, development and growth. However, most mammals, including humans, are incapable of synthesizing n-6 and n-3 HUFAs. Fish can convert C18 unsaturated fatty acids into n-6 and n-3 HUFAs via fatty acid desaturase (Fads), in which Fads2 is a key enzyme in HUFA biosynthesis. The allo-tetraploid common carp theoretically encode two duplicated *fads2* genes. The expression patterns and desaturase functions of these two homologous genes are still unknown. In this study, the full length of the *fads2a* and *fads2b* were identified in common carp (*Cyprinus carpio*). Expression analyses indicate that both genes were mainly expressed in the liver and the expression of *fads2b* is higher than *fads2a* at different developmental stages in carp embryos. Heterogenous expression and 3D docking analyses suggested that Fads2b demonstrated stronger ∆6 and ∆5 desaturase activities than Fads2a. The core promotor regions of *fads2a* and *fads2b* were characterized and found to have different potential transcriptional binding sites. These results revealed the same desaturase functions, but different activities of two homologues of *fasd2* genes in common carp. The data showed that *fads2b* played a more important role in HUFA synthesis through both expression and functional analyses.

## 1. Introduction

Highly unsaturated fatty acids (HUFAs) are fatty acids which contain three or more double bonds and longer than 20 carbon atoms. HUFAs have long been recognized as nutritional elements and essential dietary components in human health [1]. Nutritional deficiencies in n-6 and n-3 HUFAs may have adverse effects on the development of brain and neural systems, and may also lead to negative metabolic changes [2]. Dietary n-3 HUFA deficiency is associated with a potential increased risk for attention deficit hyperactivity disorder (ADHD) and other similar behavioral disorders [3]. As they are not endogenously produced by humans [4], fish have proven to be an important source of dietary sources of HUFAs [5]. The biosynthesis of HUFAs in fish involves the sequential desaturation and elongation of precursor C18 polyunsaturated fatty acids. Among fish species, there is still wide variation in the ability to synthesize HUFAs, which depends on their content of fatty acid desaturase (Fads) and elongase enzymes [6].

In fish, Fads controls the degree of unsaturation of HUFAs and catalyzes the first desaturation step [7]. More specifically, Δ6 and Δ5 desaturase catalyze the desaturation at the Δ6 and Δ5 positions in the carbon backbone during the synthesis of arachidonic acid (ARA, C20:4n-6) from linoleic acid (LA, C18:2n-6) and eicosapentaenoic acid (EPA, C20:5n-3) from linolenic acid (LNA, C18:3n-3), respectively [8]. Consequently, the biosynthetic ability of fish to produce HUFAs depends on the expression levels and activities of the aforementioned key enzymes. Among them, Δ6 desaturase is the rate-limiting enzyme, which is responsible for catalyzing the first step in the HUFA biosynthetic pathway, converting LA and LNA into γ-linolenic acid (GLA, C18:3n-6) and moroctic acid (C18:4n-3), respectively. They are also involved in the synthesis of docosahexaenoic acid (DHA, C22:6n-3) from EPA and have been commonly regarded as indicators of HUFA biosynthetic ability in fish [9]. Most studies have clearly shown that Δ6 desaturase is modulated by nutrition and environment in freshwater fish. This modulation allows for the control of lipid metabolism and the maintenance of cell membrane functionality. The Δ6 enzyme activity is stimulated at low temperatures and low salinities, and retains cell membrane fluidity. The Δ5 enzyme is another key enzyme involved in the synthesis of fish HUFAs, for which the substrates are C20:3n-6 and C20:4n-3. Molecular data has shown that *Sparus aurata* and *Oncorhynchus mykiss* Δ6 desaturase has some Δ5 activity in a yeast expression system. The enzymological data from earlier cell studies suggests that *Oncorhynchus mykiss* must have an alternate Δ5 desaturase enzyme, if the cell line truly reflects the in vivo environment in *Psetta maximus* [10]. DHA, which plays an important physiological role in vertebrates, is thought to be synthesized by Δ6 desaturation. However, the Δ4 pathway represents an alternative route for DHA biosynthesis in some teleosts. The existence of ∆4 desaturases in their genomes potentially enables them to further operate via the ∆4 pathway [11]. To investigate the transcriptional regulation of HUFA biosynthesis in fish, the upstream sequences from *fads* genes have been cloned as candidate promoters in teletosts like *Salmo salar* L. [12], *Gadus morhua* L. [12], *Lateolabrax japonicus* [13] and *Dicentrarchus labrax* [9]. Some transcription factors, like SP1 (stimulatory protein 1) [12], C/EBP (CCAAT/enhancer binding protein) [12] and NF-1 (nuclear factor 1) [14] were shown to have the binding sites in the core promoters of ∆6/∆4 desaturases genes. In comparison to freshwater fish, marine fish mainly accumulate HUFAs from phytoplankton, seaweeds and other marine aquatic organisms downstream of their food chain [15]. However, HUFAs in freshwater fish are primarily endogenously synthesized. This indicates that freshwater fish have a stronger ability to synthesize HUFAs than marine fish.

Common carp is the most suitable aquaculture species for cultivars using genomic resources for variety improvement. *Cyprinus carpio* is commonly regarded as an allo-heterotetraploid teleost fish due to its high chromosome number and DNA content and has unique genome-wide replication ability. This has led to the hypothesis that common carp may have a more complicated HUFA biosynthesis regulation mechanism than other freshwater fish [16]. Two Δ6 desaturase-like genes (Δ6*fads-a* and Δ6*fads-b*) in *Cyprinus carpio* var. Jian were cloned in a previous study. Both Δ6Fad genes were highly expressed in the liver, intestines and brain. High levels of expression were detected at early stages as early as 1 h post fertilization (hpf), but there was a significant decrease in the expression of two Δ6Fad genes from 12 hpf to 48 hpf, then a significant increase in both gene expressions from 72 hpf to 96 hpf [16,17]. Our previous work analyzed the genetic polymorphisms of the *fads2a* and *fads2b* genes in common carp, suggesting the importance of these two genes in HUFA biosynthesis and indicating that they might have different regulatory mechanisms and desaturase activities [18]. However, there is still paucity of knowledge on these two desaturase genes.

## 2. Results

### 2.1. The Differences in Gene and Protein Structures of Common Carp fads2a and fads2b

To obtain the full-length sequences of two *fads2* genes of common carp, the predicted CDS regions were amplified according to the zebrafish *fads2* gene sequence, and then the 5′ race and 3’ race ends were cloned completely from the common carp liver cDNA sample using a rapid cDNA end cloning technique. Common carp *fads2a* (GenBank accession number: MK852165.1), located in the A25 chromosome of the common carp genome (from 2,753,781 bp to 2,765,247 bp) [19], and *fads2b* (GenBank accession number: MK852166.1), located in the B25 chromosome of the common carp genome (from 4,452,255 bp to 4,465,379 bp), have a total length of 1980 bp and 1943 bp, respectively. Both genes are composed of 12 introns and 12 exons, including a 1335 bp open reading frame (ORF) (Figure 1). Although *fads2a* and *fads2b* have the same number of introns and exons and both encode 444 amino acids, the global identity between these two mRNAs was 89.10%. The protein sequence similarity between common carp Fads2a/Fads2b and zebrafish Fads2 were 90.77% and 88.51%, respectively. The similarity between common carp Fads2a and Fads2b was 89.86%. Common carp Fads2a and Fads2b, as well as zebrafish Fads2 protein sequences all have a cytochrome b5-like heme/steroid binding domain and a fatty acid desaturase domain, which indicate that they all possess a fatty acid desaturase function (Appendix A).

Predicted 3D protein structure analysis revealed that the protein structures of common carp Fads2a and Fads2b are highly similar to each other, and both are also similar to zebrafish Fasd2 (Appendix A). However, there are still 45 amino acid differences in Fads2a and Fads2b, including 6 differences in cytochrome b5-like heme/steroid binding domains and 30 differences in fatty acid desaturase domains. The protein secondary structure information showed that the Fads2a and Fads2b proteins both have 16 alpha helical structures but four and three beta pleated sheet structures, respectively (Appendix A). Further analysis revealed that there are differences in amino acid sequences in the 6th, 8th and 10th alpha helix structures of Fads2a and Fads2b, which may lead the two genes to have diverse functions.

A phylogenetic tree comparing the deduced amino acid sequences of Fads2 in common carp with those of marine fish, other freshwater fish and mammals (Appendix A) is shown in Figure 2. The analysis shows that *Cyprinus carp* Fads2 were clustered together with other two *Cyprinidae* family freshwater teleosts *Carassius auratus* and *Sinocyclocheilus anshuiensis* and were separate from marine teleosts like *Salmo salar*, which belongs to the *Salmonidae* family, in the tree. The common carp Fads2a/Fads2b protein sequences were most closely related to *S.anshuiensis* Fads2 (the similarity was 94.82% and 95.95%) and *C. auratus* Fads2 (the similarity was 95.50% and 93.47%), respectively. Combining the syntenic gene analysis between common carp and other *Cyprininae* species in our previous work [19], *fads2* genes of *Cyprinus carpio*, *Carassius auratus* and *Spinibarbus anshuiensis* might be linearly homologous and may have evolved from the same ancestor, as they all have two *fads2* genes which may be due to whole genome replication of the common ancestor.

### 2.2. Divergent Spatial and Temporal Expression Patterns of fads2a and fads2b In Vivo

Tissue distributions of *fads2a* and *fads2b* were determined by real-time quantitative PCR (qPCR) performed on cDNAs reverse transcribed from different adult (one year old) common carp organs. The results showed that *fads2a* and *fads2b* genes were predominantly expressed in the liver compared to other tissues (Figure 3a). Transcription levels of *fads2b* were significantly higher than *fads2a* in the liver (hepatopancreas) (*p* < 0.001), intestine (*p* < 0.001), brain (*p* < 0.001) and spleen (*p* < 0.05). However, the amount of expression of *fads2a* in the heart (*p* < 0.05) and gill (*p* < 0.001) were higher than *fads2b*. Although the expression patterns of *fads2a* and *fads2b* are diverse in different tissues, it can be inferred that *fads2b* plays a more dominant role in the desaturation process than *fads2a* because the synthesis of HUFAs mainly occurs in the liver.

To quantify the expression patterns of *fads2a* and *fads2b* during the early developmental stages of common carp, we analyzed their temporal expression in carp embryos at 0, 24, 48, 72 and 120 hpf by qPCR. As seen in Figure 3b, the transcript levels of *fads2b* were significantly higher than that of *fads2a* at 0, 24, 72, 96 and 120 hpf (*p* < 0.001). The expression patterns of *fads2a* and *fads2b* were both higher before the hatching period (48–72 hpf) than after. Although *fads2a* was more highly expressed than *fads2b* at 48 hpf, its expression level remained low during the larva stage (after 72 hpf) and was significantly lower than that of *fads2b* (*p* < 0.001). These results indicated that *fads2b* might play a more important role than *fads2a* during the early developmental stages of common carp.

In order to further elucidate the spatial and temporal expression of *fads2a* and *fads2b* in various common carp tissues during the early developmental stages after hatching, carp embryos were collected for whole embryo in situ hybridization at 48, 72, 96 and 120 hpf, respectively (Figure 4 and Figure 5). The results showed that the expression of *fads2a* and *fads2b* were mainly detected in the brain region at 48, 72 and 96 hpf. Notably, the transcript signals of *fads2b* (Figure 4b) were stronger than that of *fads2a* (Figure 5b) in the brain at 72 hpf. The expression of both genes were detected in the liver and intestine at 96 hpf (Figure 4c and Figure 5c) and significantly increased at 120 hpf (Figure 4d and Figure 5d). In order to study the effect of carp *fads2* genes on fatty acid content during embryonic development, the desaturase activity of these two genes was estimated by comparing the fatty acid level of C18 substrate (C18:3n-3 and C18:2n-6) with the fatty acid level of all potential desaturation products. During the whole process of embryo development, the total amount of C18 substrates decreased by nearly 50% and the content of the fads2 product showed a stable growth trend (Appendix A). In summary, the areas of expression of *fads2a* and *fads2b* are consistent in the same period during the early developmental stages of common carp, mainly in the brain and liver.

### 2.3. Differences in Desaturation Activities of Common Carp Fads2a and Fads2b in Transgenic Yeast

The desaturase functions of Fads2a and Fads2b were characterized by determining the fatty acid profiles of yeast transformed with a pYES2 vector with or without putative common carp *fads2a* and *fads2b* cDNA inserts (pYES2-Fads2a and pYES2-Fads2b). They were co-cultured with potential fatty acid substrates which included Δ6 substrates (C18:2n-6 and C18:3n-3), Δ5 substrates (C20:3n-6 and C20:4n-3), Δ8 substrates (C20:2n-6 and C20:3n-3) and Δ4 substrates (C22:4n-6 and C22:5n-3), respectively. The GC–MS analyses indicated that compared with the yeast cells transformed with pYES2 without *fads2a* and *fads2b* cDNA inserts (Appendix A), yeast cells transformed with pYES2-Fads2a and pYES2-Fads2b acquired functional HUFA desaturation activity which could effectively desaturate both Δ6 and Δ5 substrates, including C18:2n-6 (Figure 6a and Figure 7a), 18:3n-3 (Figure 6b and Figure 7b), C20:3n-6 (Figure 6c and Figure 7c) and C20:4n-3 (Figure 6d and Figure 7d). However, there was no desaturation activity detected for the Δ4 and Δ8 substrates, including C22:4n-6, C22:5n-3, C20:2n-6 and C20:20-3. Further, we analyzed the conversion rates of the Δ6, Δ5, Δ4, and Δ8 substrates to desaturated products in pYES2-Fads2a and pYES2-Fads2b transformed yeast, respectively. As shown in Table 1, the desaturation activities of Fads2b were stronger than that of Fads2a in all tested Δ6 and Δ5 substrates, including C18:2n-6, 18:3n-3, C20:3n-6 and C20:4n-3. However, both enzymes had no Δ8 and Δ4 desaturation activities. These results indicated that although common carp Fads2a and Fads2b both had the desaturated functions to Δ6 and Δ5 substrates, their activities were different.

### 2.4. Molecular Docking Analyses of Fads2a and Fads2b

To study the catalytic behavior of Fads2a and Fads2b at the molecular level during the desaturation process, the binding pockets of both Fads proteins and ten fatty acid small molecules, including the Δ6 and Δ5 substrates, were analyzed using molecular docking (Appendix A). We calculated the cavity volumes of Fads2a and Fads2b binding pockets, which were 398.2 Å^3^ and 411.7 Å^3^, respectively (Appendix A). To investigate the interaction activity between Fads2a/Fads2b and Δ6/Δ5 substrates, we specifically estimated the docking positions of the four fatty acids, including C18:2n-6, C18:3n-3, C20:3n-6 and C20:4n-3, in both Fads peptide structures (Appendix A), respectively, and the interaction energies between them (Table 2). The results of the molecular docking analysis showed that Fads2b had a lower affinity energy of interaction with C18:2n-6 and C20:3n-6 than Fads2a, which indicated that Fads2b was more active in the desaturation process than Fads2a due to its greater preference for interaction with the Δ6 and Δ5 substrates fatty acids.

### 2.5. Promoter Activity Analysis of fads2a and fads2b

In order to investigate the molecular regulation of *fads2a* and *fads2b* gene expression in common carp, sequences of different length upstream of the 5′UTR of each gene were selected to be cloned as their candidate core promoter regions. A potential promoter sequence approximately 2500 bp in length, as well as four truncated sequences of *fads2a* or *fads2b* were recombined into a pGL3-basic vector, respectively, then subsequently transfected into carp epithelioma cells (EPC). The transcriptional activity of the *fads2a* and *fads2b* gene promoters were analyzed by performing a dual luciferase test. As shown in Figure 8, five different candidate promoters for the *fads2a* (Figure 8a) and *fads2b* (Figure 8b) genes showed no significant differences in luciferase activity, which leads to the conclusion that the core active promoters of the *fads2a* and *fads2b* genes were 622 bp and 812 bp upstream of 5′UTR of each gene, respectively. The core promoter activities of *fads2b* (−812 bp to −1 bp) and *fads2a* (−622 bp to −46 bp) were also compared. The result indicated that the *fads2b* promoter showed significantly higher activity than the *fads2a* promoter (*p* < 0.001) (Figure 8c).

Finally, we predicted the potential transcription factor binding sites in the core promoters of common carp *fads2a* and *fads2b*. The results showed that TBP, NF-1 and AP-1 may regulate *fads2a*, while SP1, NF-1 and AP-1 may regulate *fads2b* (Figure 8d). The promoter regions of both genes also have the same binding site of transcription factor NF-1. It is speculated that the promoter activity of the *fads2a* and *fads2b* genes may be different due to the differences in promoter region sequences and regulatory factors.

## 3. Discussion

Fish have proven to be a unique and rich source of HUFAs [5]. Common carp, rich in HUFAs [20,21], is the most abundant global aquatic product and accounts for 10% of all freshwater production [22,23]. As a polyploid fish, possessing twice as many genes as other fish, common carp is an ideal model to study the evolution of the vertebrate polyploid genome [24]. Previous studies have successfully cloned one carp *fads* gene [10]. However, whole genome sequencing has revealed that carp have two *fads* genes rather than one. This discovery highlights the need for more in-depth research into the study of desaturase functions in the HUFA’s synthesis pathways of common carp.

In contrast to mammals, which have distinct *fads1* and *fads2* genes with separate specificities of Δ5 and Δ6 activities [25], the *fads1* gene has been lost during evolution in fish. This phenomenon may be caused by a comprehensive process of gene replication, gene loss, and functional diversity [26,27]. As a result, most desaturation steps in the HUFA biosynthetic pathway are catalyzed by Fads2 enzymes in fish. In this study, the full-length sequences of the common carp *fads2a* and *fads2b* genes were successfully cloned for the first time. The gene structure of two *fads2* genes were fundamentally the same, including 12 introns and 12 exons, and both encoded 444 amino acids. However, the amino acid sequence similarity of the Fads2a and Fads2b genes is 89.86%, which shows a higher degree of difference than the gene structure. Three-dimensional structure analysis showed differences in positions and lengths of alpha helix and beta pleated sheet structures, which indicated that the *fads2a* and *fads2b* genes might also have different desaturation activities, functions and regulation mechanisms in the process of unsaturated fatty acid synthesis.

Expression studies of the *fads2a* and *fads2b* genes in different adult common carp tissues revealed both genes were expressed in all experimental tissues, while expression levels in various tissues were different. Both the *fads2a* and *fads2b* genes exhibited the highest expression in liver and brain tissues, which are the main effector organs of desaturase. However, the expression of *fads2b* is higher than that of *fads2a* in those tissues. Based on these results, it is speculated that *fads2b* plays a stronger role and is more efficient in the process of HUFAs synthesis than *fads2a*.

Temporal and spatial expression analysis of *fads2a* and *fads2b* by whole embryo in situ hybridization revealed that the transcripts of *fads2* and *fads2b* were maternally expressed, as well as appearing in different expression patterns over time. The *Fads2b* gene was expressed at 24 hpf and showed an increased expression trend in the subsequent development period, which led to speculation that it might play an important role throughout the entire embryonic development stage in common carp. Comparatively, the expression of the *fads2a* gene remained at low levels before 72 hpf and increased significantly at 96 hpf and 120 hpf. This result indicated that the *fads2a* gene mainly regulates the metabolism of fatty acids and participates in the synthesis of HUFA when carp embryos develop into fry after membrane emergence.

The study of spatial expression of *fads2a* and *fads2b* in various tissues during common carp embryonic development demonstrated that the transcripts of both genes had positive signals in the brain regions at a very early stage. Comparatively positive signals develop in the early embryonic stage. In comparison, the expression of *fads2a* increased sharply and gained functionality after the carp embryo emerged from the membrane. In adult common carp, the expression of *fads2b* in most tissues is higher than that of fads2a, suggesting that the *fads2b* gene plays a major role in the HUFA synthesis pathway.

The desaturation activities of common carp Fads2a and Fads2b were investigated in transgenic yeast in this study. The results showed that both Fads2a and Fads2b had ∆5 and ∆6 activities. However, the activities of ∆4 and ∆8 were not verified. Experimentally, transfection of *Saccharomyces cerevisiae* functioned to verify gene expression in vitro, but also required secondary verification in vivo. Unlike freshwater fish like common carp, ∆4 desaturase activity is mostly found in seawater fish, such as *Siganus canaliculatus* [28], *Trachinotus ovatus* [29], *Leuresthes sardina*, etc. The addition of exogenous fatty acids can inhibit the metabolic pathway of ∆6 activity. Studies have shown that Atlantic salmon fed with fish oils rich in EPA and DHA have significantly lower HUFA biosynthesis pathway activity in the liver and intestines compared to those fed with fish oil that did not contain EPA and DHA [30], indicating that the level of n-3 series unsaturated fatty acids directly affects HUFAs that were found in the liver after 96 hpf. These results suggest that the *fads2a* and *fads2b* gene are mainly expressed in the liver after hatching.

In the process of common carp embryonic development, the total amount of C18 fatty acids substrate (C18:3n-3 and C18:2n-6) decreased by nearly 50%. DHA content increased steadily during the stage from 0 hpf to 120 hpf. Combined with the results of temporal and spatial expression of *fads2a* and *fads2b*, it is speculated that *fads2b*, which is expressed in the head of carp embryo, would synthesize DHA in the brain and contribute to brain development. The *Fads* gene combined with ∆6 activity plays a certain regulatory role in the synthesis of HUFAs. The expression of ∆6 *fads* mRNA is lower in the liver of Atlantic salmon fed with fish oil than those fed with the vegetable oil control [31].

We defined the active promoter regions of common carp *fads2a* and *fads2b* in this study. We then screened the predicted transcription factors, TBP and NF-1 for *fads2a,* NF-1, Sp1 and AP-1 for *fads2b*. Sp1 is particularly important in constitutive expression of the Atlantic salmon ∆6 *fads* gene. ∆6 *fads* has lower promoter activity and tissue expression levels in Atlantic cod than Atlantic salmon and other seawater fish [32]. This difference may be caused by the absence of Sp1 binding sites in the promoter region of Atlantic cod ∆ 6 *fads* [12]. As a ubiquitous universal transcription factor, Sp1 has also been confirmed in the activation of the human ∆6 *fads* gene promoter [33]. Compared with the *fads2b* promoter, the promoter of *fads2a* shows lower activity and does not contain Sp1 binding sites. The results of the promoter activity comparison were consistent with the results of the yeast transfection experiment in vitro. The conversion rate of Fads2b to different products was also higher than that of Fads2a. Therefore, it can be inferred from this study that the difference between common carp *fads2a* and *fads2b* may be due to the existence of SP1 binding sites in the *fads2b* gene promoter, which are necessary for the complete expression of ∆6 activity of the common carp *fads* gene.

Our study successfully cloned two *fads2* genes of common carp and revealed that both *fads2a* and *fads2b* had ∆6/∆5 desaturation activities and had a similar expression pattern. However, the expression level, desaturation activities and promoter activities of *fads2a* were generally higher than *fads2b*.

## 4. Materials and Methods

### 4.1. Animals and Ethics Statement

Common carp studied in this research were cultivated for one year in the breeding base at the Chinese Academy of Fishery Sciences (Beijing, China), with the same commercial diet. Common carp samples were collected for cloning and quantitative expression analysis of the *fads2a* and *fads2b* genes. The experimental common carp were one year old, and the body weight was 500 g on average. Common carp embryos studied in this research were obtained by artificial fertilization and incubated in 25 °C fresh water with an oxygen supply. This research was approved by the committee for the ethics of animal experiments of the Chinese Academy of Fishery Science.

### 4.2. Sequence and Structure Analysis of Two Common Carp fads2 Genes

To identify the common carp *fads2* gene, the zebrafish *fads2* mRNA sequence (GenBank accession number: NM_131645.2) was aligned to the recent common carp genome (GenBank accession number: GCA_018340385.1) [19]. Two homologous genome locations which overlapped with the whole zebrafish *fads2* cDNA sequences were selected as candidate common carp *fads2* genes. The protein sequences and complete open reading frames (ORFs) of these two candidate genes were predicted by FGENESH+ web server (http://www.softberry.com (accessed on 28 October 2019)) by taking the zebrafish Fasd2 sequence (GenBank accession number: AAH49438.1) as the template. To verify the ORF sequences of the two common carp *fads2* genes, specific primers were designed for 5′ and 3′ rapid amplification of cDNA ends (RACE) to obtain full-length transcripts (Appendix A). To compare the three-dimensional structures of the common carp and zebrafish Fads2 proteins, their amino acid sequences were used for ab initio modeling with the I-TASSER server [34]. The modeled protein structures were refined through molecular dynamics simulations with AMBER16 software [35].

### 4.3. Phylogenetic Analysis

The protein sequences of common carp Fads2a and Fads2b aligned with selected mammalian and other defined teleost Δ6 desaturase proteins (Appendix A) were used for constructing the phylogenetic tree by the neighbor-joining method of MEGA7 [36,37]. Confidence of tree branch topology was obtained by a bootstrapping measurement of 1000 iterations [38]. Finally, the phylogenetic tree was embellished using the iTOL v6.6 web server (https://itol.embl.de/itol.cgi (accessed on 24 October 2022)) [39].

### 4.4. GC–MS Analysis of Common Carp Embryos at Different Developmental Stages

Common carp embryos at 0, 24, 48, 72, 96 and 120 hpf were collected during development and freeze-dried after grinding into a powder while adding liquid nitrogen into the mortar. Fatty acids methyl esters (FAMEs) were prepared with 2 mL of 0.5 M NaOH/MeOH at 100 °C for 30 min after ultrasonic concussion for 10 min, followed by the addition of 2 mL 14% BF3·MeOH (wt/vol) at 100 °C for 1 h. FAMEs were extracted with 1 mL of hexane and 5 mL of a saturated solution of NaCl by vigorous mixing then centrifugation at 1000× *g* for 5 min. The aqueous top phase was transferred to a clean glass tube and the lower phase was used to extract FAMEs a second time by adding 1 mL hexane. The two FAME extracts were combined and dried in a stream of oxygen-free nitrogen. The FAMEs were then resuspended in 1 mL of hexane and filtered through a nylon syringe filter SCAA-104 (ANPEL, Shanghai, China). Fatty acid analyses were performed using a 7890A GC System (Agilent Technologies, Santa Clara, CA, USA) equipped with a flame ionization detector. The gas chromatograph was equipped with a capillary column (60 m × 0.25 mm i.d., Agilent Technologies, Santa Clara, CA, USA) hydrogen was used as the carrier gas. Samples were applied when the temperature gradient was from 60 to 150 °C at 10 °C/min, to 200 °C at 15 °C/min and finally to 230 °C at 30 °C/min. Individual methyl esters were identified by comparing the GC retention times with GLC NESTLE 37 MIX (NU-CHEK, Elysian, MN, USA), methyl cis-7,10,13,16-docosatetraenoate, methyl cis-7,10,13,16,19-docosapentaenoate and methyl cis-4,7,10,13,16,19-docosahexaenoic (ANPEL, Shanghai, China). Meanwhile, FA was quantitatively analyzed by adding methyl nonadecanoate (C19:0) as the internal standard. The FA content of common carp embryos were measured as ((C19:0 content/C19:0 area) × (FA area/dehydration embryo weight)).

### 4.5. Heterologous Expression of fads2 ORFs in Yeast

ORF fragments of two common carp *fads2* cDNAs were amplified from common carp liver cDNA with primers containing restriction sites for Hind III and Xho I (Appendix A). The amplified DNA products were constructed into a pYES2 vector (Invitrogen, Waltham, MA, USA). The constructed plasmids, pYES2-Fads2a and pYES2-Fads2b, were extracted using an AxyPrep Plasmid Miniprep Kit (Axygen, Union City, CA, USA) and transformed into *S. cerevisiae* (strain INVSc1, a gift form Nanhai institution, Chinese Academy of Fishery Sciences) by using the S. c. EasyComp™ Transformation Kit (Invitrogen, Waltham, MA, USA), respectively.

The transformed yeast were grown in *S. cerevisiae* minimal medium-uracil overnight at 30 °C, then the cultures were diluted to an OD600 of 0.4 for further growth. When the yeast cultures reached an OD600 of 1, 2% (wt/vol) galactose was used for protein induction and supplemented with different fatty acid (FA) substrates at final concentrations of 0.5 mM (C18), 0.75 mM (C20) and 1 mM (C22), respectively. After 48 h, the yeast was centrifuged and washed twice, then dried by vacuum freeze-drying for 12 h. FAME preparation and FA analysis of yeast samples were performed as described above for common carp embryos. The conversion rate of potential substrates was calculated by the proportion of substrate FA converted to desaturated products as (product area/(product area + substrate area)) × 1000.

### 4.6. Expression of fads2 Genes during Common Carp Ontogeny

The fish were anaesthetized using a eugenol solution at a concentration of 40 mg/L. Twelve tissues including the liver, intestine, brain, muscle, eye, heart, spleen, skin, pancreas, gill, blood and kidney were sampled from each fish and immediately collected in 1.5 mL RNase-free centrifuge tubes, frozen in liquid nitrogen then stored in a −80 °C freezer until being further processed. Embryos at 0, 24, 48, 72, 96 and 120 hpf were sampled for RNA isolation. Expression of the common carp fads genes was measured by qPCR. The qPCR analysis was performed on a 7500 Real Time PCR System (Thermo Fisher Scientific, Waltham, MA, USA) using SYBR Green Realtime PCR Master Mix (TOYOBO, Japan) according to a previously described procedure [40]. The relative expression of the target gene was normalized with β-actin expression calculated by the 2^−∆∆Ct^ method. Primers used for qPCR are shown in Appendix A.

### 4.7. Whole-Mount In Situ Hybridization (WISH)

Digoxygenin-labelled RNA probes of common carp *fads2a* and *fads2b* were synthesized in vitro. First, approximate 400 bp *fads2a* and *fads2b* DNA templates were cloned from the common carp liver cDNA using EasyTaq PCR Supermix kit (TransGen Biotech, Beijing, China), 1 μL common carp liver cDNA, 25 μL 2× EasyTaq Supermix, 1 μL WISH-F/WISH-R primers (Appendix A) and 22 μL ddH_2_O. They were processed in a PCR reaction program as follows: initial denaturation at 94 °C for 4 min, 35 cycles of denaturation at 94 °C for 20s—primer annealing at 60 °C for 20s—extension at 72 °C for 30s, and final extension at 72 °C for 5 min. Second, the products were inserted into the pEASY-T3 cloning vector (TransGen Biotech, Beijing, China) by setting up the following cloning reaction program: 0.5 μL PCR products, 1 μL pEASY-T3 cloning vector and 3.5 μL ddH_2_O were incubated at 25 °C for 5 min. Third, positive clones were identified by colony PCR using M13 forward and reverse primers and inoculating positive clones in Amp+ LB liquid medium. Then, plasmid DNA was isolated using a plasmid DNA MiniPrep Kit (Thermo Scientific, Waltham, MA, USA). The identified DNA templates of *fads2a* and *fads2b* RNA probes were analyzed by sequencing with M13 forward and reverse primers. Finally, RNA probes were transcribed the from 1μg purified DNA template by using the MAXIscript™ T7 transcription kit (Thermo Scientific, Waltham, MA, USA), 2 μL 10× transcription buffer, 2 μL DIG-Labling mix, 1 μL RNase inhibitor, and 1 μL RNA polymerase mix with up to 20 μL DEPC water were incubated for 3 h at 37 °C.

The 48, 72, 96 and 120 hpf common carp embryos were collected and fixed in 4% paraformaldehyde (PFA). Pigmentation of embryos was removed in 3% H_2_O_2_/0.5% KOH and dehydrated in an ascending MeOH row as described previously [41]. In brief, embryos were re-hydrated in a descending row of MeOH, washed in PBST and permeabilized using 10 μg/mL proteinase K at room temperature (20 min for 48 hpf embryos, 50min for 72 hpf embryos, 70 min for 96 hpf embryos and 90 min for 120 hpf embryos). Digoxigenin (DIG)-labeled RNA probes were generated using DIG RNA Labeling Mix (Roche, Germany) with a MAXIscript kit (Invitrogen, Waltham, MA, USA) as described above. Whole-mount in situ hybridization (WISH) was performed as described previously [41].

### 4.8. Molecular Docking

The crystal structure for common carp Fads2a and Fads2b were unavailable in the PDB databank, thus the homology models of these two proteins were ab initio constructed using I-TASSAR software [42]. The best three-dimensional (3D) structure of Fads2a/Fads2b by homology modeling was further optimized using AMBER16 software [43]. The structures of four fatty acid substrates (C18:2n-6, C18:3n-3, C20:3n-6 and C20:4n-3) used for molecular docking in this research were obtained from Pubchem database (https://pubchem.ncbi.nlm.nih.gov (accessed on 19 September 2019)). Binding of fatty acid substrates to Fads2a/Fads2b were investigated by molecular docking using the AutoDock (version 4.2), and the docking complexes with highest binding energy were used in the following molecular dynamics (MD) simulations using AMBER16 software.

### 4.9. Cloning of fads2a/fads2b Promoters and Construction of Expression Plasmids

The *fads2a/fads2b* candidate promoter region sequences were obtained from genomic sequencing data of common carp. To identify the core promoters of common carp *fads2a*/*fads2b*, different length (576 bp, 971 bp, 1484 bp, 1895 bp, 2479 bp length of *fads2a* upstream sequences and 811 bp, 1257 bp, 1786 bp, 2227 bp, 2675 bp length of *fads2b* upstream sequences) candidate promoter regions from −2525 to −46 bp upstream *fads2a* gene and −2676 to −1 bp upstream *fads2b* gene was amplified from common carp genomic DNA by PCR. The forward primers with the Xho I restriction endonuclease site and the reverse primers with the Hind III restriction endonuclease site are shown in Appendix A. Then, different length products were purified by AxyPrepTM PCR Cleanup Kit (Axygen Biosciences, Union City, CA, USA) and constructed into a linearized pGL3-Basic vector (Promega, Madison, WI, USA) by T4 DNA ligase (Takara, Shiga, Japan) overnight at room temperature.

### 4.10. Cell Culture, Transfection and Luciferase Assay

Transient transfections were performed through lipofection using Lipofectamine 2000™ reagent (Invitrogen, Waltham, MA, USA) in the presence of reduced serum medium (Invitrogen, Waltham, MA, USA) according to the manufacturer’s instructions when EPC cells reached 90–95% confluency, 0.8 µg of pGL3 luciferase reporter vectors (Promega Corporation, Madison, WI, USA) were used for a co-transfection with 0.08 µg of the pRL-CMV vector (Promega Corporation, Madison, WI, USA) for luciferase analysis normalization. Transfected cells were incubated at 28 °C without CO_2_.

At 48 h post transfection, EPC cell extracts were collected and firefly and Renilla luciferase activities were measured using a Dual Glo Luciferase Assay System (Promega Corporation, Madison, WI, USA). Briefly, 75 mL of the remaining DMEM serum-free medium was mixed with 75 mL of Dual-Glo™ Luciferase Reagent (Promega Corporation, Madison, WI, USA) and incubated for 1 min. The lysates were measured for firefly luciferase activity in a 96-well microplate-reading luminometer (Veritas™ Microplate Luminometer, Promega Corporation, Madison, WI, USA). Each sample was normalized to the absorbance of Renilla luciferase to correct for variations in transfection efficiency using 75 mL of Stop & Glo^®^ Reagent (Promega Corporation, Madison, WI, USA) added to the same well and incubated for 10 min before detecting. Experiments were performed in duplicate.

### 4.11. Statistical Analysis

All data were analyzed using GraphPad Prism software (GraphPad, San Diego, CA, USA). We performed an independent *t*-test to examine whether the expressions of *fads2a* and *fads2b* genes were significantly different in each tissue or each hour post-fertilization we measured in the qPCR analyses. Asterisks denote statistical significance (* *p* < 0.05; ** *p* < 0.01; *** *p* < 0.001). All data are reported as mean ± SD. Means and standard deviations are from at least four independent experiments.

## Figures and Tables

**Figure 1 ijms-24-10638-f001:**
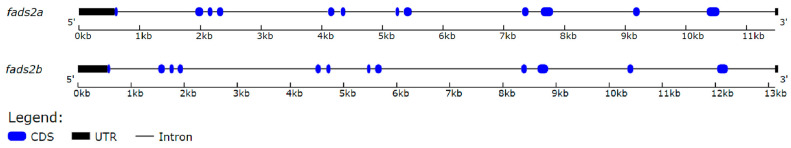
Gene structure of common carp fads2a (MK852165.1) and fads2b (MK852166.1) were determined from the genome sequence from the NCBI database. CDS regions are indicated by blue boxes. UTR regions are indicated by black boxes. Non-coding intron regions are indicated by black lines. The gene structure display was illustrated by Gene Structure Display Server (GSDS) 2.0.

**Figure 2 ijms-24-10638-f002:**
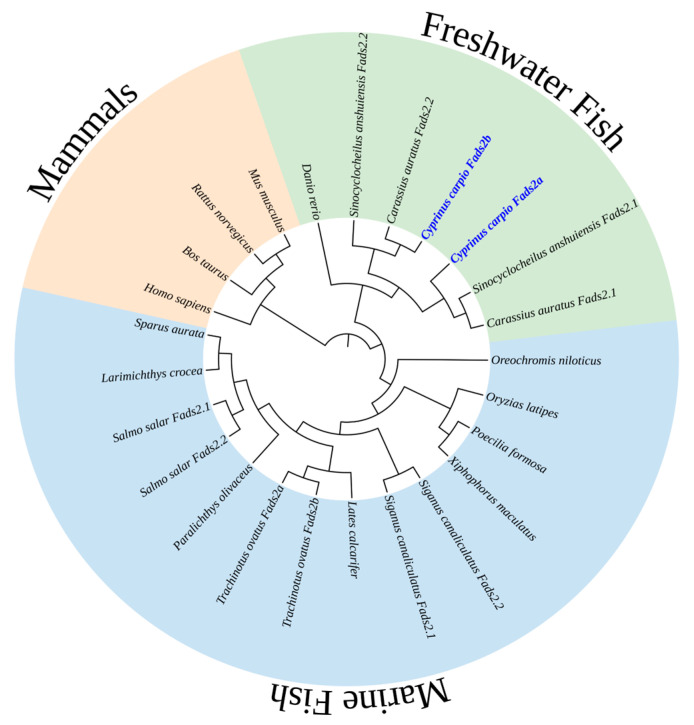
Phylogenetic tree comparing the amino acid sequences of common carp (*Cyprinus carpio*) Fads2a and Fads2b (labeled by blue color) with desaturase proteins from other aquatic organisms and mammals.

**Figure 3 ijms-24-10638-f003:**
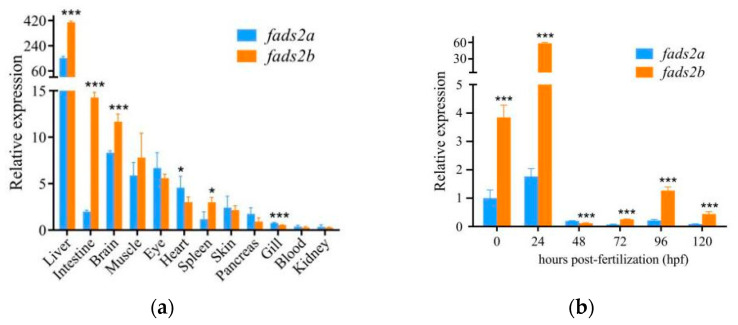
(**a**) Tissue expression profiles of fads2a and fads2b in one-year old common carp. The results are reported as the mean ± standard deviation, *n* = 4, * *p* < 0.05, *** *p* < 0.001. (**b**) qRCR analyses of the temporal expression patterns of fads2a and fads2b during common carp embryogenesis. The results are reported as the mean ± standard deviation, *n* = 4, *** *p* < 0.001.

**Figure 4 ijms-24-10638-f004:**
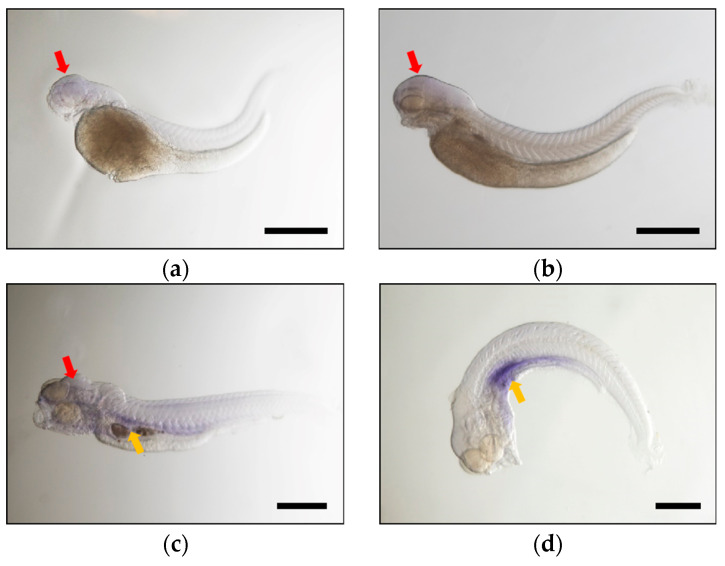
Whole mount in situ hybridization showing the expression of *fads2a* at 48 hpf (**a**), 72 hpf (**b**), 96 hpf (**c**), and 120 hpf (**d**) during common carp embryogenesis. The *fads2a* transcript signals were labeled with blue dye. Black scale bars: 500 μm. The red arrow indicates the head region, the liver and intestine are marked with the yellow arrow.

**Figure 5 ijms-24-10638-f005:**
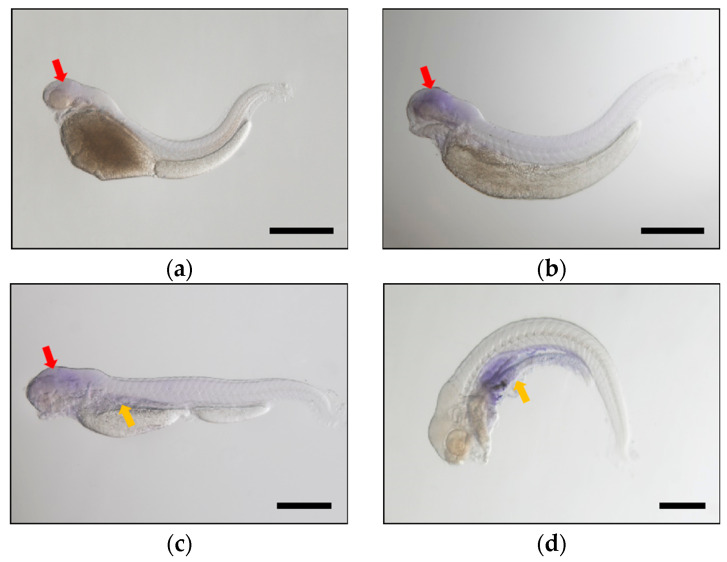
Whole mount in situ hybridization showing the expression of *fads2b* at 48 hpf (**a**), 72 hpf (**b**), 96 hpf (**c**), and 120 hpf (**d**) during common carp embryogenesis. The *fads2b* transcript signals were labeled with blue dye. Black scale bars: 500 μm. The red arrow indicates the head region, the liver and intestine are marked with the yellow arrow.

**Figure 6 ijms-24-10638-f006:**
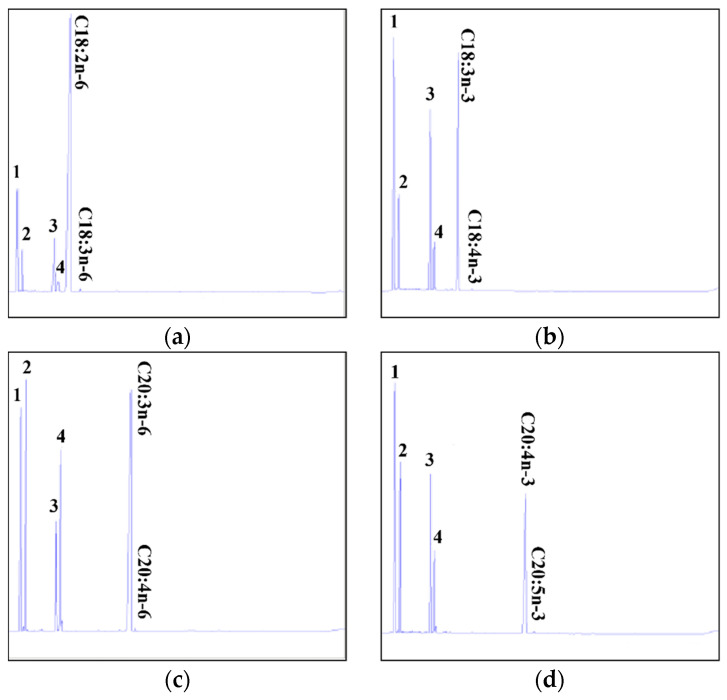
Characterization of common carp Fads2a desaturase activities in transgenic yeast (*S. cerevisiae*) grown in the presence of fatty acid substrates C18:2n-6 (**a**), C18:3n-3 (**b**), C20:3n-6 (**c**) and C20:4n-3 (**d**). According to the retention time of each fatty acid by GC analysis, C18:3n-6 (**a**), C18:4n-3 (**b**), C20:4n-6 (**c**), C20:5n-3 (**d**) are indicated to be the unsaturated fatty acid products. Peaks 1 to 4 represent C16:0 (1), C16:1n-7 (2), C18:0 (3) and C18:1n-9 (4), respectively, which are the four main endogenous fatty acids of *S. cerevisiae*. The vertical axis displayed the flame-ionization detector response peak. The horizontal axis showed the peak retention time (min).

**Figure 7 ijms-24-10638-f007:**
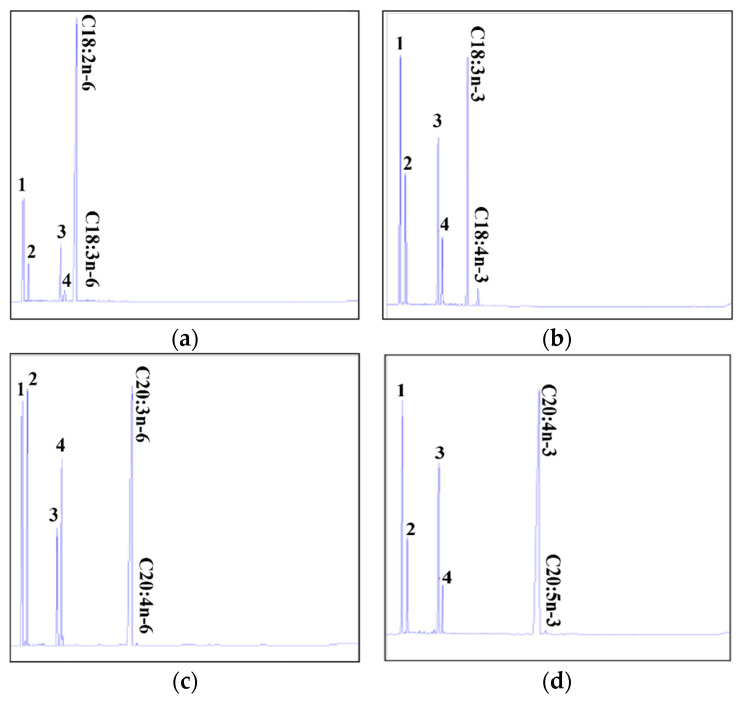
Characterization of common carp Fads2b desaturase activities in transgenic yeast (*S. cerevisiae*) grown in the presence of fatty acid substrates C18:2n-6 (**a**), C18:3n-3 (**b**), C20:3n-6 (**c**) and C20:4n-3 (**d**). According to the retention time of each fatty acid by GC analysis, C18:3n-6 (**a**), C18:4n-3 (**b**), C20:4n-6 (**c**), C20:5n-3 (**d**) are indicated to be the unsaturated fatty acid products. Peaks 1 to 4 represent C16:0 (1), C16:1n-7 (2), C18:0 (3) and C18:1n-9 (4), respectively, which are the four main endogenous fatty acids of *S. cerevisiae*. The vertical axis displayed the flame-ionization detector response peak. The horizontal axis showed the peak retention time (min).

**Figure 8 ijms-24-10638-f008:**
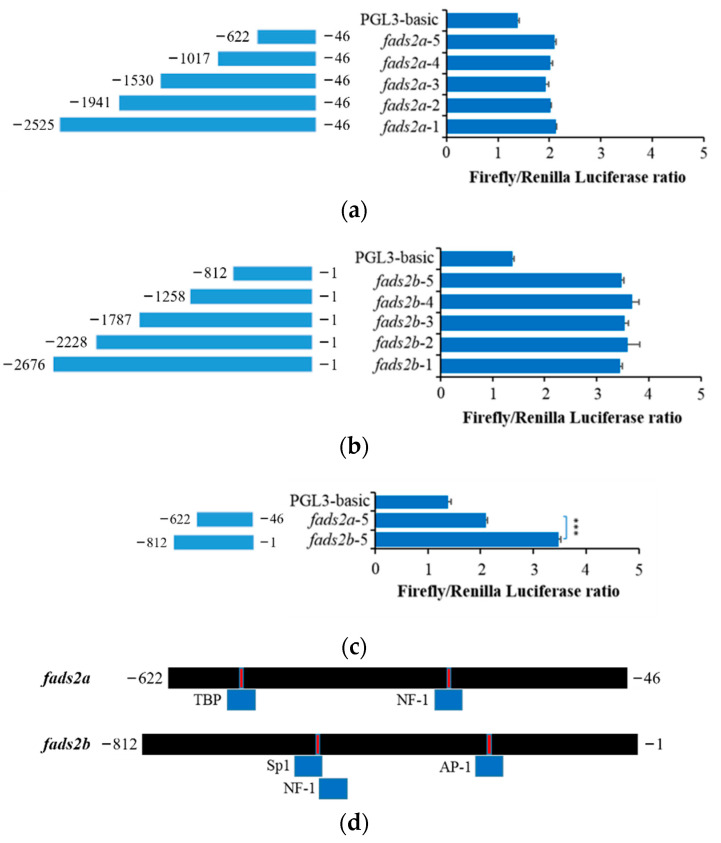
Promoter activity analysis of common carp *fads2a* (**a**) and *fads2b* (**b**). The results are reported as the mean ± standard deviation, *n* = 5, *** *p* < 0.001. (**c**) Core promoter activity analysis between common carp *fads2a* and *fads2b* gene. (**d**) Transcription factor prediction of common carp *fads2a* and *fads2b*.

**Table 1 ijms-24-10638-t001:** Functional characterization of common carp Fads2a and Fads2b.

	FA Substrate	Product	FA Substrate Peak Area (pA·s)	Product Peak Area (pA·s)	Conversion (%)
Fads2a	C18:2n-6	C18:3n-6	30,945.8	101.4	0.33
Fads2a	C18:3n-3	C18:4n-3	325.5	4.8	1.45
Fads2a	C20:3n-6	C20:4n-6	3817.5	10.9	0.28
Fads2a	C20:4n-3	C20:5n-3	2194.9	17.2	0.78
Fads2b	C18:2n-6	C18:3n-6	8603.6	51.4	0.59
Fads2b	C18:3n-3	C18:4n-3	7850.4	318.2	3.89
Fads2b	C20:3n-6	C20:4n-6	2762.5	8.9	0.32
Fads2b	C20:4n-3	C20:5n-3	3870	42	1.07

Results are expressed as the conversion rates of ∆6, ∆5, ∆8, and ∆4 substrates to desaturated products in pYES2-Fads2a and pYES2-Fads2b transformed yeast.

**Table 2 ijms-24-10638-t002:** Molecular docking energy analysis of Fads2a and Fads2b.

Fatty Acids	Docking Energy (kcal/mol)	
	Fads2a	Fads2b
C18:2n-6	−5.86	−5.04
C18:3n-3	−5.83	−6.13
C20:3n-6	−5.98	−5.54
C20:4n-3	−5.73	−5.84

## Data Availability

The data presented in this study are available in the Appendix A.

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
