# Peer review of "Fads2b Plays a Dominant Role in ∆6/∆5 Desaturation Activities Compared with Fads2a in Common Carp (Cyprinus carpio)"

_ijms, 2023, doi:10.3390/ijms241310638_

Round 1
Reviewer 1 Report
It is a good quality MS. Insignificant typos were found in Lines 131-132.
My advice is also to rewrite the sentence in Lines 279-280 and cite Castro et al., 2012. Marked in the attached MS.

Insignificant typos were found in Lines 131-132.
My advice is also to rewrite the sentence in Lines 279-280 and cite Castro et al., 2012. Marked in the attached MS.
Author Response
Reply to the comments of reviewer 1 point by point:
Minor comments:
Question 1: Insignificant typos were found in Lines 131-132.
Reply: We corrected these two names to “Salmo salar” (Line 132 of revised manuscript) and “C. auratus” (Lines 134-135 of revised manuscript).
Question 2: Advice to rewrite the sentence in Lines 279-280 and cite Castro et al., 2012. Marked in the attached MS.
Reply: We followed your suggestion and add a sentence of “This phenomenon may be caused by a comprehensive process of gene replication, gene loss, and functional diversity.” (Lines 281-282 of revised manuscript) and two references as following:
- Castro, L.F., et al., Functional desaturase Fads1 (Delta5) and Fads2 (Delta6) orthologues evolved before the origin of jawed vertebrates. PLoS One 2012, 7,(2): p. e31950.DOI: 10.1371/journal.pone.0031950.
- Lopes-Marques, M., et al., Retention of fatty acyl desaturase 1 (fads1) in Elopomorpha and Cyclostomata provides novel insights into the evolution of long-chain polyunsaturated fatty acid biosynthesis in vertebrates. BMC Evol Biol 2018, 18,(1): p. 157.DOI: 10.1186/s12862-018-1271-5.

Reviewer 2 Report
I think you should define HUFA, as not common. More than ... double bonds and longer than...
line 33 in human health or healthy human , correct pls
line 34 brain and neural system
line 40 which depends on
line 41 replace composition with content or activity
line 49 and forward, do not repeat the short name explanations of fatty acids, already mentioned
line 52 definition of 22:6 is lacking
line 63 remove definition of 22:6
line 72 shown to have
line 84 remove significant : High level of....
line 187 replace content with level
line 307 fry is both singular and plural , remove fries !!!!!!
it is an interesting MS, some English is not perfect but most is understandable.
Author Response
Reply to the comments of reviewer 2 point by point:
Comments and Suggestions for Authors:
Question 1: I think you should define HUFA, as not common. More than ... double bonds and longer than...
Reply: We added a define of HUFA as “Highly unsaturated fatty acids (HUFAs) are fatty acids which contain three or more double bonds and longer than 20 carbon atoms.” (Lines 32-33 of revised manuscript).
Question 2: line 33 in human health or healthy human , correct pls.
Reply: We corrected it. (Line 34 of revised manuscript).
Question 3: line 34 brain and neural system.
Reply: We corrected it. (Lines 35-36 of revised manuscript).
Question 4: line 40 which depends on.
Reply: We corrected it. (Line 42 of revised manuscript).
Question 5: line 41 replace composition with content or activity.
Reply: We corrected it to “content” (Line 43 of revised manuscript).
Question 6: line 49 and forward, do not repeat the short name explanations of fatty acids, already mentioned.
Reply: We deleted the repeated short name explanations of LA and LNA (Line 52 of revised manuscript).
Question 7: line 52 definition of 22:6 is lacking.
Reply: We added the short name explanation of DHA (Lines 53-54 of revised manuscript).
Question 8: line 63 remove definition of 22:6.
Reply: We removed it (Line 64 of revised manuscript).
Question 9: line 72 shown to have.
Reply: We corrected it (Lines 72-73 of revised manuscript).
Question 10: line 84 remove significant : High level of....
Reply: We corrected it (Line 85 of revised manuscript).
Question 11: line 187 replace content with level.
Reply: We corrected it (Line 188 of revised manuscript).
Question 12: line 307 fry is both singular and plural , remove fries !!!!!!
Reply: We corrected it (Line 308 of revised manuscript).

Reviewer 3 Report
The study aimed to investigate the Δ6/Δ5 desaturation activities of Fads2a and Fads2b in common carp.
The paper is well structured and well written and contains all relevant information for answering the questions for the study's purposes.
There are only minor, mainly editorial modification requires.
Those are in order of appearance in the text as follows:
L 33: health instead of healthy
L 34: n-6 and n-3 HUFAs have other effects than neural development; however, they are the most important.
L 57: retain cell membrane fluidity instead of retain cell membranes fluids
L 60: Please, explain „injection studies”.
L 63: Please, add the Latin name of turbot, such as in other species.
L 73-74: What about carnivorous fish species if marine fishes mainly accumulate HUFAs from phytoplankton and seaweeds?
L 130: Please, refer here that both species belong to the Cyprinidae family.
L 131: Please, refer here that the cause of the difference is that Salmo salar belongs to the Salmonidae family.
Figure 2: Mammals instead of Mammalian (please modify to mammals in the figure legend)
L 149: Common carp has no liver but hepatopancreas; however, it can be using the term liver, but it would be preferable to add hepatopancreas in brackets.
L 189: Please, refer here to which C18 fatty acid precursors content decreased by 50% because the table data only partly prove this statement.
L 191: The statement: „similar tissue distribution patterns during the early developmental stages” requires a more detailed explanation because there is only in situ hybridisation data did not quantify.
L 313: The statement „fads2b plays a key role in the early stage of embryonic development” did not proven by the results because in situ hybridisation data did not quantify.
L 321-322: Scientific names of fishes would be preferable.
L 324: fish oils rich in EPA and DHA instead of EPA and DHA fish oils
L 325-326: Please, modify this sentence because the cited reference did not contain „fish oil that didn’t contain HUFAs.
L 329-330: Please, double-check the data in the supplementary table.
L 357-361: In the subchapter ”4.1. Animals and ethics statement,” please, add some details about keeping conditions of one-year-old carps and embryos.
L 439-440: Please, add a relevant reference to the statement „according to a previously described procedure.”
Author Response
Reply to the comments of reviewer 3 point by point:
Minor comments:
Question 1: L 33: health instead of healthy.
Reply: We corrected it (Line 34 of revised manuscript).
Question 2: L 34: n-6 and n-3 HUFAs have other effects than neural development; however, they are the most important.
Reply: We added more effects of HUFAs (Line 36 of revised manuscript) and one more reference as following:
- Liput, K.P., et al., Effects of Dietary n-3 and n-6 Polyunsaturated Fatty Acids in Inflammation and Cancerogenesis. Int J Mol Sci 2021, 22,(13).DOI: 10.3390/ijms22136965.
Question 3: L 57: retain cell membrane fluidity instead of retain cell membranes fluids.
Reply: We corrected it (Lines 58-59 of revised manuscript).
Question 4: L 60: Please, explain „injection studies”.
Reply: We deleted this description (Line 62 of revised manuscript).
Question 5: L 63: Please, add the Latin name of turbot, such as in other species.
Reply: We corrected it (Line 64 of revised manuscript).
Question 6: L 73-74: What about carnivorous fish species if marine fishes mainly accumulate HUFAs from phytoplankton and seaweeds?
Reply: We added “other marine aquatic organisms downstream of their food chain” (Lines 74-75 of revised manuscript).
Question 7: L 130: Please, refer here that both species belong to the Cyprinidae family.
Reply: We added this information (Line 131 of revised manuscript).
Question 8: L 131: Please, refer here that the cause of the difference is that Salmo salar belongs to the Salmonidae family.
Reply: We added this information (Lines 132-133 of revised manuscript).
Question 9: Figure 2: Mammals instead of Mammalian (please modify to mammals in the figure legend)
Reply: We edited the Figure 2 as following and corrected the figure legend (Line 143 of revised manuscript).
Question 10: L 149: Common carp has no liver but hepatopancreas; however, it can be using the term liver, but it would be preferable to add hepatopancreas in brackets.
Reply: We added it (Line 150 of revised manuscript).
Question 11: L 189: Please, refer here to which C18 fatty acid precursors content decreased by 50% because the table data only partly prove this statement.
Reply: We corrected “C18 precursors” to “C18 substrates” (Line 150 of revised manuscript) which included C18:3n-3 and C18:2n-6 (identified in Line 188 of revised manuscript).
Question 12: L 191: The statement: „similar tissue distribution patterns during the early developmental stages” requires a more detailed explanation because there is only in situ hybridisation data did not quantify.
Reply: We rewrote this statement as “In summary, the areas of expression of fads2a and fads2b are consistent in the same period during the early developmental stages of common carp, mainly in the brain and liver.” (Lines 191-193 of revised manuscript).
Question 13: L 313: The statement „fads2b plays a key role in the early stage of embryonic development” did not proven by the results because in situ hybridisation data did not quantify.
Reply: We deleted this statement (Line 313 of revised manuscript).
Question 14: L 321-322: Scientific names of fishes would be preferable.
Reply: We corrected these three names (Lines 321-322 of revised manuscript).
Question 15: L 324: fish oils rich in EPA and DHA instead of EPA and DHA fish oils.
Reply: We corrected it (Line 324 of revised manuscript).
Question 16: L 325-326: Please, modify this sentence because the cited reference did not contain „fish oil that didn’t contain HUFAs.
Reply: We corrected “fish oil that didn’t contain HUFAs” to “fish oil that didn’t contain EPA and DHA” based on the cited reference (Lines 325-326 of revised manuscript).
Question 17: L 329-330: Please, double-check the data in the supplementary table.
Reply: We changed “C18 fatty acids (as substrates)” to “C18 fatty acids substrate (C18:3n-3 and C18:2n-6)” (Lines 329-330 of revised manuscript).
Question 18: L 357-361: In the subchapter ”4.1. Animals and ethics statement,” please, add some details about keeping conditions of one-year-old carps and embryos.
Reply: We added more information (Lines 358-360, 362-364 of revised manuscript).
Question 19: L 439-440: Please, add a relevant reference to the statement „according to a previously described procedure.”
Reply: We added a relevant reference here (Line 443 of revised manuscript) as following:
- Zhao, R., et al., Dominant Elongase Activity of Elovl5a but Higher Expression of Elovl5b in Common Carp (Cyprinus carpio). Int J Mol Sci 2022, 23,(23).DOI: 10.3390/ijms232314666.
